# Compressive Reconstruction Based on Sparse Autoencoder Network Prior for Single-Pixel Imaging

**Hong Zeng [1], Jiawei Dong [2,3], Qianxi Li [2,3], Weining Chen [2], Sen Dong [2], Huinan Guo [2] and Hao Wang [2,\*]**

[1] DFH Satellite Co., Ltd., Beijing 100094, China; zenghong1980@163.com
[2] Xi'an Institute of Optics and Precision Mechanics, Chinese Academy of Sciences, Xi'an 710119, China; dongjiawei22@mails.ucas.ac.cn (J.D.); liqianxi22@mails.ucas.ac.cn (Q.L.); cwn@opt.ac.cn (W.C.); dongsen@opt.ac.cn (S.D.); guohuinan@opt.ac.cn (H.G.)
[3] University of Chinese Academy of Sciences, Beijing 100049, China
\* Correspondence: wanghao@opt.ac.cn

**Abstract:** The combination of single-pixel imaging and single photon-counting technology enables ultra-high-sensitivity photon-counting imaging. In order to shorten the reconstruction time of single-photon counting, the algorithm of compressed sensing is used to reconstruct the underdetermined image. Compressed sensing theory based on prior constraints provides a solution that can achieve stable and high-quality reconstruction, while the prior information generated by the network may overfit the feature extraction and increase the burden of the system. In this paper, we propose a novel sparse autoencoder network prior for the reconstruction of the single-pixel imaging, and we also propose the idea of multi-channel prior, using the fully connected layer to construct the sparse autoencoder network. Then, take the network training results as prior information and use the numerical gradient descent method to solve underdetermined linear equations. The experimental results indicate that this sparse autoencoder network prior for the single-photon counting compressed images reconstruction has the ability to outperform the traditional one-norm prior, effectively improving the reconstruction quality.

**Keywords:** sparse autoencoder network prior; single-photon counting compressive imaging; single-pixel imaging; multi-channel prior; numerical gradient descent

## 1. Introduction

Single-pixel imaging, also known as computational ghost imaging, is an imaging method based on compressed sensing. It loads a series of patterns on the spatial light modulator (SLM) to modulate the imaging scene, using a point detector without spatial resolution to detect the correlation intensity and then using the measured light intensity value and the measurement matrix corresponding to the pattern to restore the image [1]. Single-pixel imaging and single-photon counting technology can be combined to realize single-photon counting compression imaging if a single-photon detector is used. It has two main advantages. One is that two-dimensional imaging can be achieved only by using point detectors. Compared with the imaging method using single-photon avalanche photodiode array, multi-anode photomultiplier tube, microchannel plate photomultiplier tube, and other array single-photon detectors, it has lower cost and higher resolution, especially in infrared, Terahertz, and other special frequency bands [2,3]. Secondly, the point detector in the single-pixel imaging system can collect the light intensity of multiple pixels at the same time. By utilizing the relationship between photon count values and light intensity, the imaging sensitivity of the system is no longer limited by the detection sensitivity of the single-photon point detector, and thus, the imaging sensitivity of the system can be very high. By using a single-photon detector with photon limit sensitivity as point detection, imaging sensitivity can be further improved so that single-photon counting imaging that exceeds the single-photon limit can be achieved [4]. Therefore, single-photon counting

compression imaging will have broad application prospects in ultra-weak light imaging detection, such as medical diagnosis, astronomical observation, and spectral measurement.

For a two-dimensional single-photon image, the observation value $f \in \mathbb{R}^r$ can be modeled as follows:

$$f = A\bar{x} + w, \tag{1}$$

where $A \in \mathbb{R}^{r \times mn}$ is a linear operator, and $w \in \mathbb{R}^r$ is a zero-mean Gaussian white noise. The problem of image reconstruction is to extract $x$ from $f$ through a linear system $Ax = f$. However, under normal circumstances, the system is underdetermined ($r < mn$) or ill-conditioned (such as deconvolution and deblurring). The classic least squares method will cause noise amplification, blurring, and overlap, so it is no longer applicable.

In order to stabilize the reconstruction, additional constraints and prior knowledge are required. In the early stage, the development of single-photon image reconstruction methods mainly came from the regularization constraints based on compressed sensing theory [5] and the introduction of explicit priors [6], such as one-norm prior [7], Bayesian prior [8], etc. The use of sparsity and low-rank prior [9] information can quickly reconstruct images at low sampling rates, and good reconstruction effects can be obtained. The one-norm prior is a structured sparsity constraint that mainly contributes to feature extraction and selection, making the training more efficient and interpretable. The L2-norm can also be used as a regularization term, preventing overfitting and improving the model's generalization ability, but it cannot make the network sparse so that it does not meet the sparsity requirements of compressed sensing. Therefore, the previous work often uses one-norm as the prior information. The method using sparsity and low-rank prior, known as TVAL3, is really competitive in compressed sensing, particularly performing well at a high sampling rate, which effectively solves a class of equality-constrained non-smooth optimization problems with a particular structure. While this method is limited by the sampling rate, its performance drops sharply when the sampling rate decreases.

Recently, neural networks have been used to assist rapid imaging and general image reconstruction problems. They can be roughly divided into two methods. One method is to use a traditional matrix with fixed acquisition methods and the elements that can be changed in a specific way (such as Gaussian matrix, Hadamard matrix, etc.) as measurement matrix to realize the linear measurement of the image, and the neural network is used to solve it. This method can avoid a large number of calculations brought by traditional iterative algorithms so as to achieve rapid reconstruction [10]. Another method is to use neural networks for joint learning sampling and reconstruction. The first layer of the network uses a fully connected layer, which eliminates bias and activation functions, and its weight matrix is used as the measurement matrix for compressed sensing after training. Since the measurement matrix is obtained through joint optimization with the reconstruction part, this method not only has a faster reconstruction speed but also has a better reconstruction quality than the iterative algorithm [11]. However, the number of weights of the fully connected layer will increase exponentially with the dimension of the reconstructed image. When the reconstructed image is very large, the reconstruction burden will be greatly increased.

This paper proposes a novel sparse autoencoder network prior for single-photon image reconstruction combined with compressed sensing theory. Due to the powerful feature extraction capability of the sparse autoencoder network and the superiority of the network prior, the experimental results demonstrate that the reconstruction performance of this method is better than that of the traditional one-norm prior. Our contributions can be summarized as follows:

- We proposed a novel compressed sensing reconstruction method with a sparse autoencoder network prior that can be directly applied in photon-counting compressed imaging systems. Compared with the traditional one-norm prior, this sparse autoencoder network prior has significant advantages in terms of reconstruction quality;
- We proposed the concept of multi-channel prior information, and our experiments demonstrated that reconstructing images under the constraints of multi-channel net-

work priors could effectively solve the problems encountered with single-channel network priors. Good reconstruction results can be achieved regardless of the low or high measurement rate.

## 2. Compressive Reconstruction System for the Single-Pixel Imaging

A compressive reconstruction system based on the sparse autoencoder network prior for the single-pixel imaging is shown in Figure 1. The combination of single-pixel imaging and single-photon counting technology was proposed by Wenkai Yu et al. in 2012 [12]. The parallel light generator is composed of an LED light source with an output power of 10 W, a collimator, an attenuator, and a diaphragm. The LED light source emits parallel light through the collimator, attenuator, and diaphragm, making the intensity of parallel light at the single-photon level, which illuminates the imaging target and then, the image on the Digital Micro-Mirror Device (DMD) through a convex lens. The DMD is composed of 1024 × 768 micro-mirror arrays. Each micro-mirror can control the rotation angle of the micro-mirror through a binary matrix loaded on the DMD to realize spatial light modulation. In the binary matrix, "1" corresponds to the rotation of the micro-mirror +12°, and "0" corresponds to the rotation of the micro-mirror −12°. We use a single-photon detector (PMT) to collect the reflected photons with a deflection of +12° and transmit them into a specially developed FPGA-based single-photon pulse counting circuit. The PMT is used in a photon counting mode. In this mode, the incident light is weak so that the photomultiplier tube outputs a discrete sequence of single-photon pulses, where one pulse represents the detection of a photon. The density of a single-photon pulse represents the intensity of light, and they are directly proportional. For this reason, we could obtain the accurate light intensity without considering the instability probability caused by current and voltage. The circuit loads the generated measurement matrix onto the DMD and, at the same time, counts the single-photon pulses from the PMT synchronously to obtain the photon pulse count value and then combines the prior information obtained after the sparse autoencoder network training to reconstruct the final image.

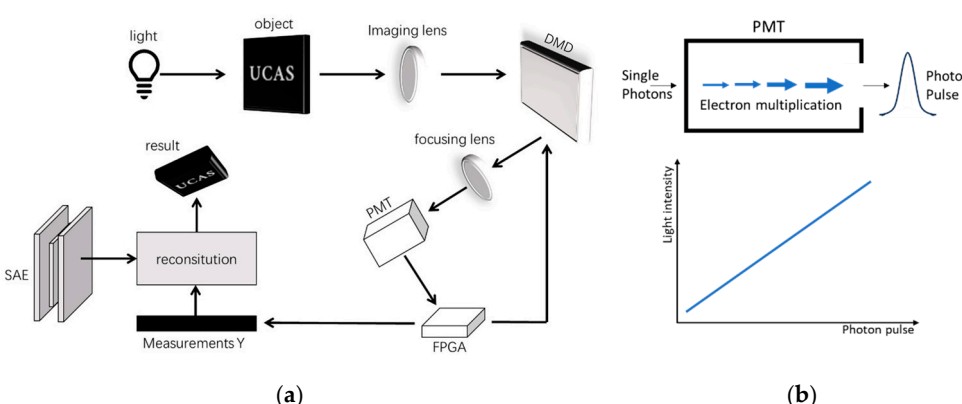

(**a**)  (**b**)

**Figure 1.** The compressive reconstruction system we used for single-pixel imaging. (**a**) The structure of the system. The object is imaged on the DMD after the illumination by the parallel light and then collected by PMT. The signal processed by FPGA is the measurement value Y, which is used to reconstruct the picture with the prior information. (**b**) The relationship between photon pulse and light intensity.

The traditional prior information reconstruction method uses a random matrix (such as a Gaussian random matrix) that meets the properties of RIP as the measurement matrix and uses an iterative algorithm based on compressed sensing plus a prior regular term to reconstruct a clear image. For example, the OMP algorithm uses a compressed sensing method based on a norm prior to iteratively reconstruct the image [8,13]. Under the premise that the measurement matrix is fixed and the hardware quality is not considered, the quality of image reconstruction depends on the selection of the prior information and

the pros and cons of the iterative algorithm. Traditional one-norm prior can no longer meet the requirements of today's image reconstruction. For images with obvious characteristics, we propose a new reconstruction method based on a sparse autoencoder network prior, which effectively improves the reconstruction effect of one-norm prior. This will be introduced in detail in Section 3.

## 3. Sparse Autoencoder Network Prior-Based Reconstruction Method

### 3.1. Compressed Sensing

Compressed sensing, also known as compressive sampling or sparse sampling, is a technique for finding sparse solutions to underdetermined linear systems. The image reconstruction process of solving the underdetermined equation $y = Ax$ is based on known measurement values $y$ and measurement matrix $A$ to obtain the original image $x$. It is widely used in single-photon compression imaging technology [14]. Therefore, single-photon compression imaging technology based on compressed sensing methods heavily relies on the inherent sparsity of single-photon images. If we define a manifold prior as $prior(x)$, the image reconstruction model can be expressed as follows:

$$\underset{x}{Min}||Ax - y||^2 + \lambda \times prior(x), \tag{2}$$

where $x \in \mathbb{R}^{mn}$ is the reconstructed image; $A \in \mathbb{R}^{r \times mn}$ represents a partially sampled random matrix; $y \in \mathbb{R}^r$ represents the original data obtained in the single-photon imaging system, and $\lambda$ is a hyperparameter that balances the effects of image fidelity and prior constraints.

### 3.2. Sparse Autoencoder Network

Sparse autoencoder [15] is an unsupervised learning algorithm used for learning feature representation. It is a type of autoencoder neural network that transforms input data into a series of encoding values and reconstructs them back to the original inputs using a decoder. Its main target is to extract important features from high-dimensional data sets by learning a low-dimensional representation. During this process, sparse autoencoder usually imposes sparsity constraints on encoding values to make the learned features more robust and interpretable. A standard sparse autoencoder model typically consists of three parts: encoder; decoder; and loss function. Figure 2 shows a diagram of a simple sparse autoencoder.

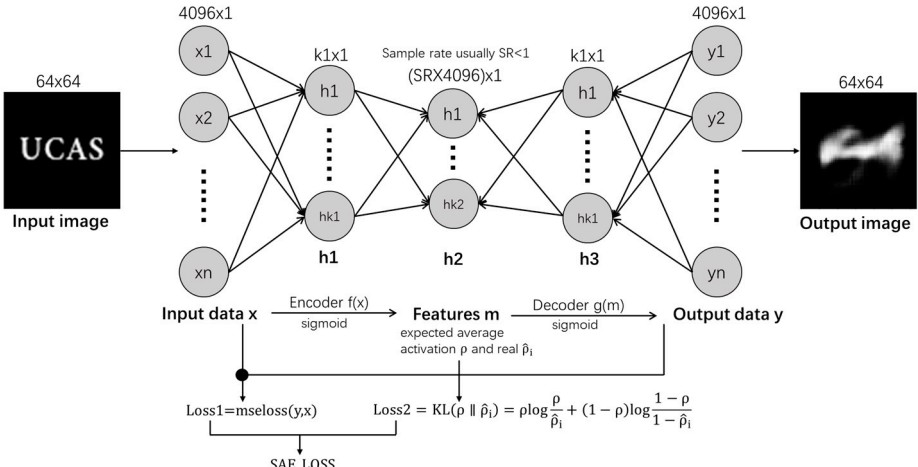

**Figure 2.** The principle and structure of sparse autoencoder. The input image is reshaped to a column vector; xn is the input data, representing n-dimensional features of the input. h1, h2, and h3 are the hidden layers, which means the extraction and reconstruction of image features. The SAE_LOSS is the loss function we use in this experiment, consisting of two parts, MSELoss and sparsity penalty.

In its specific structure, the original image UCAS image is given as input; then, the image is first transformed into a low-dimensional feature vector through the encoder. The encoder consists of fully connected layers, which compress the low-dimensional feature vector in a feature-specific manner. Then, the compressed feature vector is fed into the hidden layer, and the activation function of the hidden layer is usually selected as Sigmoid or ReLU. In the hidden layer, regularization of the activation value is necessary to handle issues caused by insufficient activation leading to overfitting. The activation value is adjusted to the sparsity constraint, and a sparse vector is output to the decoder. When this sparse vector reaches the decoder, it is projected onto a fully connected output layer first, which is equal in size to the pixels of the image. Then, the output of this layer will enter the reversal embedding operation, which remaps the high-dimensional feature vector compressed by the encoder back to the original input space and is used for reconstructing the original image. The feature extraction diagram is shown in Figure 3.

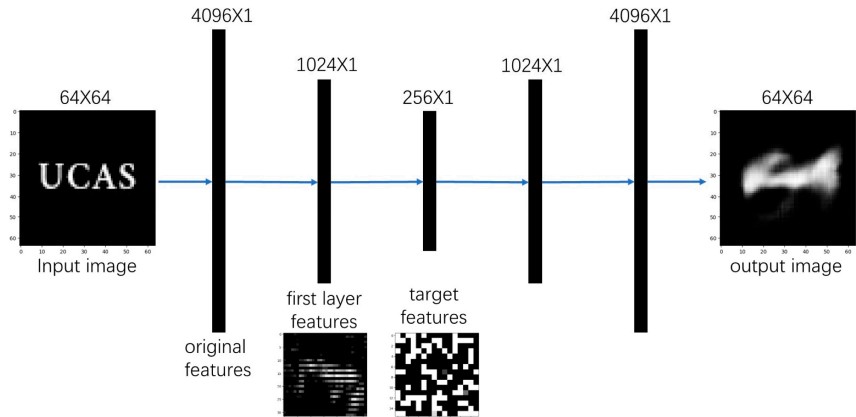

**Figure 3.** The feature extraction diagram of SAE network. We use two down-sampling layers to obtain the texture features and visualize the features through shape transformation.

In a sparse autoencoder, if the dimension of the hidden layer is greater than or equal to the input dimension, its parameters will simply store the input data and output them when required. Although this method achieves high accuracy during training, the neural network will experience overfitting when mapping the data identically. To prevent the autoencoder network from mechanically copying the input to the output, it is necessary to learn an under-complete representation of the input that forces it to capture the most relevant features in the training data. This approach is often used to extract the most useful features from the input signal.

Throughout the encoding and decoding process, to control the activity level of neurons in the hidden layer, the sparse autoencoder introduces a sparsity constraint term in the loss function, which constrains the activation status of the hidden layer to obtain better feature abstractions and sparse representations. The loss function of the sparse autoencoder consists of two key parts: reconstruction error and sparsity penalty. The reconstruction error, named MSE loss, is mainly used to constrain the error of the model's decoder when reconstructing the sample, while the sparsity penalty forces the encoder model to learn encoding values that conform to a certain sparsity level. This model can optimize the weights of the autoencoder based on the reconstruction error, while sparse regularization can be achieved by restricting the weights of the encoder or adding a penalty term to the loss function. The sparsity coefficient is a hyperparameter in the network that determines how much non-zero encoding should be learned. A low sparsity coefficient means that the network can learn more non-zero encoding to be more adaptive to noise.

Finally, the model training is completed by optimizing the loss function, including the reconstruction error from the decoder output and the sparsity constraint from the hidden layer output. This approach allows for the gradual optimization of the parameters of the

encoder and decoder, enabling the autoencoder to learn the characteristics of input data and better reconstruct input images.

In image reconstruction, sparse autoencoder has certain advantages compared to other neural networks such as dual fully connected layer autoencoder and U-Net. Firstly, when the activation rate of the intermediate hidden layer of the sparse autoencoder is limited, it not only increases the compression degree of the image but also improves the modeling of local rules and the effect of eliminating image noise. Secondly, sparse autoencoder has fewer parameters and consumes fewer resources, making them easier to train and infer. Moreover, the feature vectors learned by sparse autoencoder are interpretable and better suited for dimensionality reduction processing of noisy images. Finally, in terms of application range, sparse autoencoders have been successfully applied in various image processing applications and perform better in traditional tasks such as low-noise, denoising, and dimensionality reduction compression.

In choosing the sparse autoencoder as a prior constraint for image reconstruction, this decision is rooted in prior work where Alain et al. [16] established a connection between the output of a sparse autoencoder network, denoted as $D_x(u)$, and the true data density $p(u)$, as described by the following Equation (3).

$$D_x(u) = \frac{\int (u - \eta) g_{\sigma_\eta}(\eta) p(u - \eta) d\eta}{\int g_{\sigma_\eta}(\eta) p(u - \eta) d\eta}, \tag{3}$$

In the equation above, $g_{\sigma_\eta}(\eta)$ is a Gaussian kernel with a standard deviation of $\sigma_\eta$. This kernel is a smooth function characterized by rotational symmetry and translational invariance.

As evident from this equation, the network output $D_x(u)$ is a weighted average of the image within the input neighborhood. In other words, the neural network can generate a reconstructed image that closely resembles the original image by considering the probabilities of pixel values within the input image regions, as well as the noise present in the image. This implies a smoothed and weighted relationship between the output image of the neural network and the regional pixels of the original image rather than a simple one-to-one correspondence between pixel points. This equation mathematically explains the principles underlying neural network image reconstruction and elucidates the prerequisite for employing the neural network as prior information—the neural network acting as a prior can regulate relationships among regional pixels.

Moreover, $D(x)$ progressively aligns with the input image x as the neural network loss $(D(x) - x)$ iteratively evolves. The change in $D(x)$ can be derived through simultaneous differentiation of both sides of Equation (3), resulting in Equation (4). This equation shows that the autoencoder error $D(x) - x$ is directly proportional to the smoothed gradient of the logarithmic likelihood.

$$D(x) - x = \sigma_\eta^2 \nabla log[g_{\sigma_\eta} * p](x), \tag{4}$$

This equation implies that when there is a significant variation in the region of $p(x)$, particularly when this region contains texture information, $D(x) - x$ should also exhibit substantial changes. As the autoencoder undergoes successive training iterations, the error $D(x) - x$ approaches a minimum at either local or global extrema. This indicates that within the context of image fidelity term reconstruction in this study, the proposed prior constraint can gradually guide the fidelity term $Ax - y$ toward an optimal output image that adheres to texture features. This verifies that this prior constraint facilitates the incorporation of texture information into the $D(x) - x$ iterative training process.

Fundamentally, this sparse autoencoder network's prior constraint resembles the principle of structured sparsity constraint, specifically the one-norm prior constraint. The one-norm prior sets the probabilities of insignificantly contributing pixels to zero, reducing noise and artifacts. Similarly, the sparse autoencoder network's prior constraint regulates based on the pixel grayscale variations within regions, aiding image reconstruction. Both

approaches impact reconstruction at the pixel level. Consequently, the squared autoencoder error can be used as a prior to influence image reconstruction. This motivation has led to the proposal of a single-photon compressed imaging method based on the prior constraint of the sparse autoencoder network.

In order to obtain better prior information from the SAE network, we optimized the training learning rate, network parameters, and loss function parameters, enabling the SAE prior network to better assist in image reconstruction.

### 3.3. Single-Pixel Imaging Based on the Sparse Autoencoder Network Prior

The experimental procedure is shown in Figure 4. We can obtain the reconstruction of the image by solving the following objective function:

$$J(x) = \underset{x}{Min}||Ax - y||^2 + \lambda \times ||D(x) - x||^2, \tag{5}$$

where $x \in \mathbb{R}^{mn}$ is the image to be reconstructed; $A \in \mathbb{R}^{r \times mn}$ represents a partially sampled random matrix; $y \in \mathbb{R}^r$ represents the original data obtained in the single-photon imaging system; $D(x)$ is the output of the network, and $\lambda$ is a hyperparameter, which balances the compressed sensing fidelity term and the prior constraint of the sparse autoencoder network influences.

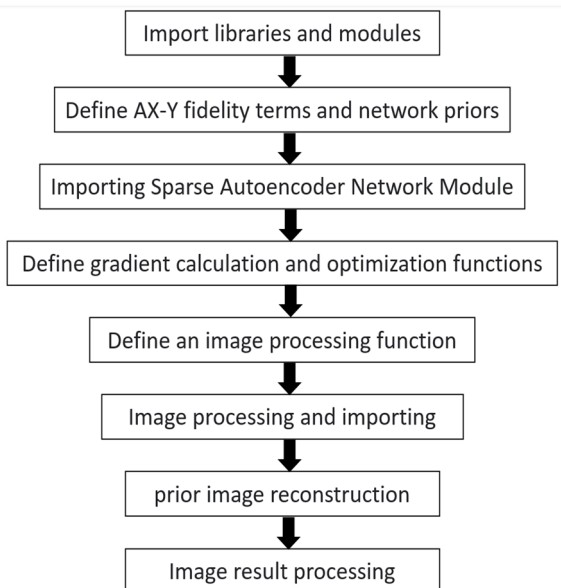

**Figure 4.** The experimental procedure shows the detailed design process of this experiment.

The entire reconstruction process can be divided into two steps: training the sparse autoencoder network to obtain prior information $prior(x)$ and solving the objective function to reconstruct the image. The main difference from the classical iterative method is that during the image reconstruction process, we apply a novel prior constraint based on the sparse autoencoder network in addition to the compressive sensing fidelity term. This constraint will result in more desirable results compared to the traditional one-norm prior of the input image. The method of solving this objective function will be discussed in detail later.

The success of the sparse autoencoder prior reconstruction method mainly lies in two aspects:

1.  The superiority of the network prior. It has been demonstrated in earlier works that neural networks themselves are a form of prior knowledge, and their different structures limit their ability to learn information. The image restoration ability constrained by the prior of the network is superior to many state-of-the-art, non-local, patch-based

priors, such as the BM3D prior [17]. We replace one-norm prior with SAE prior to change the role of the prior from a sparsity constraint to a contour similarity constraint, making this process more interpretable;

2. The powerful feature extraction ability of the sparse autoencoder network allows it to capture the most significant features in the training data. Using these features as prior information for image restoration is advantageous in obtaining more accurate results.

Apart from the single-channel network priors described above, the reconstruction performance can be improved by increasing the number of prior information channels, as shown in Figure 5. We combine multiple different prior information as constraints for image reconstruction and assign them certain weights of influence. Through experiments, we reach the conclusion that single-channel network priors often have their limitations. For example, the reconstruction effect at low sampling rates is better than that at high sampling rates. The performance of multi-channel network priors can effectively express the advantages of each single-channel network prior while avoiding their shortcomings.

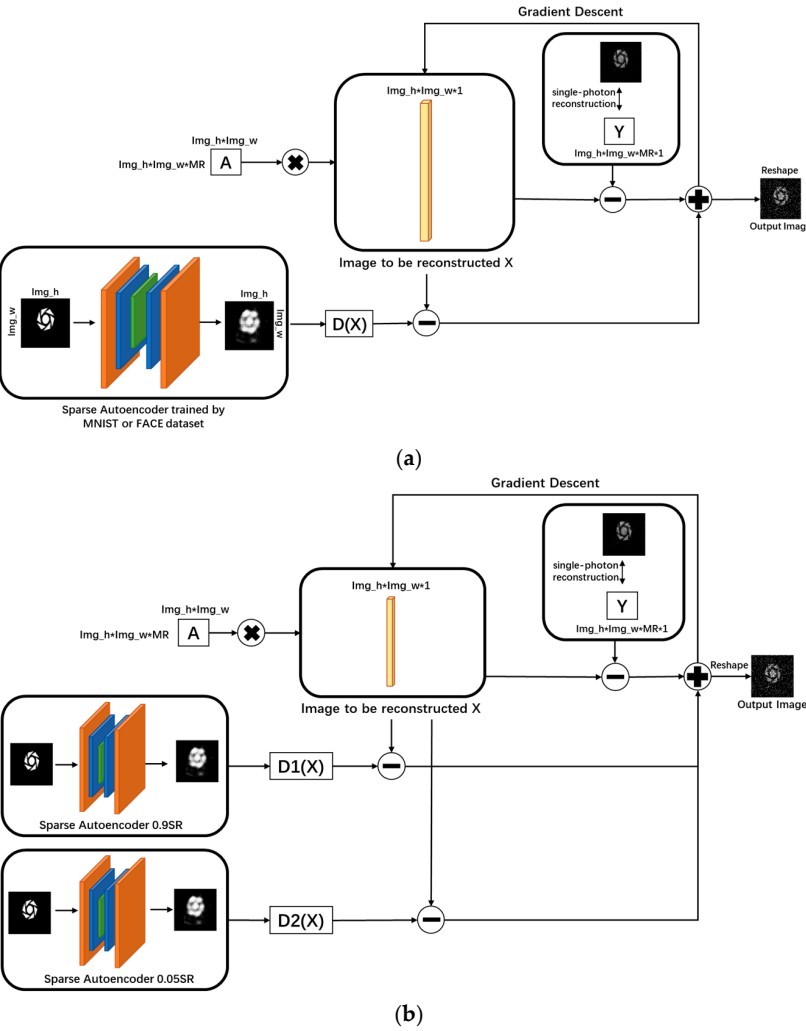

**Figure 5.** The calculation structure of single-channel and multi-channel network priors. (**a**) The structure of single-channel. We use the pre-trained SAE network to obtain the prior information D(x) and then initialize the image to be reconstructed x (img_h height and img_w width) as an all-zero vector (img_h* img_w height and 1 width). A is the measurement matrix, and Y is the measurement value obtained via this experiment. x is calculated using gradient descent method as function 5. (**b**) The structure of multi-channel. We use two different SAE sampling rates to obtain the prior information.

### 3.4. The Sparse Autoencoder Network Training and Reconstruction Method

3.4.1. The Sparse Autoencoder Network Training

Assuming that the input image is $x \in \mathbb{R}^{mn}$, $W \in \mathbb{R}^{R \times mn}$ is the weight value, and $b \in \mathbb{R}^R$ is the bias, the core operation of the fully connected (FC) layer is the product operation of the matrix vector:

$$Output = Wx + b, \tag{6}$$

Through the influence of nonlinear activation functions, its essence is a nonlinear transformation from one feature space to another. Any dimension in the target space, that is, any node in the hidden layer, is believed to be influenced by each dimension of the source space. The sparse autoencoder network using a fully connected layer can effectively perform classification learning of different features of the input image. Compared with a convolutional autoencoder network, this network is actually a convolution operation with a kernel size equal to the size of the upper layer feature. The result after convolution corresponds to a node, which is a point in the fully connected layer. The advantage lies in that it can comprehensively capture global information and capture the most significant features of the input image.

Therefore, for the above reasons, we design the sparse autoencoder network to be a simpler dual fully connected layer. The first layer reduces the dimensionality of the input image so that the dimension of the hidden layer is much smaller than that of the input image and realizes the encoding process of the input image. The second layer restores the dimensionality to that of the input image, decoding the output reconstruction result. During the training process, assuming the network output is $D(x)$, we minimize the output result to satisfy the following equation:

$$Loss_{SAE} = E_x\left[||D(x) - x||^2\right], \tag{7}$$

where $E_x\left[||D(x) - x||^2\right]$ is the expectation calculation.

3.4.2. Reconstruction Method

After obtaining the final parameters of the sparse autoencoder network training, we substitute them as the Equation (5).

This is a minimum value-solving problem with a penalty term, and there are many methods for solving it, such as the Augmented Lagrange multiplier method (ALM) [18,19], Alternating direction augmented Lagrange multiplier method (ADMM) [20], gradient descent method [21–23], and so on. Among them, the solutions of ALM and ADMM are to solve the problem by decomposing it into multiple minimum values without penalty terms, and their iteration count is very large, resulting in the reconstruction time exhibiting exponentially with the image size. Compared to the first two methods, the reconstruction time of the gradient descent method is significantly shorter, so we finally choose the gradient descent method to solve the above equation. The solving algorithm steps are shown in Algorithm 1.

---

**Algorithm 1.** The algorithm steps for solving with gradient method

---

1: Initialization: $x^0$
2: for $k = 1, 2, ..., K$ do
3: $\nabla f(x^k) \approx \frac{f(x^k+h) - f(x^k-h)}{2h}$
4: while the following equation is satisfied to stop iterating:
5: $\nabla f\left(x^k\right) < loss\_max$
6: end(while)
7: update $x^{k+1} = x^k - \nabla f(x) \times leaning\_rate$
8: end(for)
9: Output: $x^{k+1}$

---

The key to the gradient descent method is the accuracy of gradient calculation, and traditional methods of calculating gradients are usually divided into two categories: analytical and numerical. The analytical method is to find the expression of the function derivative and find the extreme value of the expression, that is, the gradient of the original function. However, the derivative expression of most functions is very difficult to solve and prone to errors. Therefore, considering the accuracy of gradient calculation, we use numerical methods to calculate the gradient by using small changes in the independent variable to estimate the derivative. Although this method is time-consuming, it provides higher accuracy.

Assuming $f(x)$ is the function of the gradient to be obtained, $h$ is the minimum value, which is set to 0.0001, using the following Taylor series expansion:

$$f(x+h) = f(x) + f'(x)h + f''(x)h^2 + O\left(h^3\right), \tag{8}$$

$$f(x-h) = f(x) - f'(x)h + f''(x)h^2 + O\left(h^3\right), \tag{9}$$

$$f(x+h) - f(x-h) = 2f'(x)h + O\left(h^3\right), \tag{10}$$

Therefore, the gradient of the function is as follows:

$$f'(x) = \frac{f(x+h) - f(x-h)}{2h} + O(h^2), \tag{11}$$

where $O\left(h^2\right)$ is the calculation error, the value is very small and can be ignored, so we think

$$f'(x) \approx \frac{f(x+h) - f(x-h)}{2h}, \tag{12}$$

## 4. Result and Discussion

### 4.1. Implementation Details

We conducted four sets of simulation experiments and one application experiment to evaluate the impact of sparse autoencoder prior on image reconstruction. In the simulation experiments, we used the MNIST dataset and FACE dataset to train fully connected sparse autoencoder networks and applied the results to reconstruct the "西光所" and "UCAS" images; we reconstructed the imaging system measurement value y with the calculated value $Ax_0$. $A$ is the measurement matrix corresponding to different measurement rates, and $x_0$ is the original image; then, we obtained the theoretical measurement values y corresponding to different MR. The process is shown in Figure 6. In the application experiments, the photon counting method is used to reflect the light intensity; then, we obtained the number of photon pulses from the system as the real measurement value $y$.

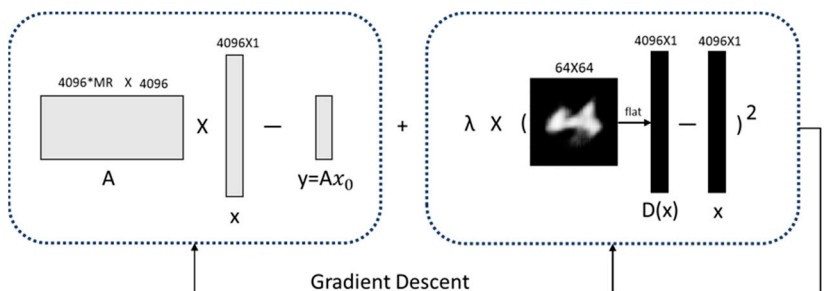

**Figure 6.** The structure of the simulations. We use the calculated y = $Ax_0$ as the measurement value, and the size of $A$ is changed along with the MR (4096*MR height and 4096 width). Then, we use the procedure mentioned in Section 3.4 to reconstruct the image.

Firstly, we fixed the network structure and changed the size of the hidden layer in the sparse autoencoder network, that is, the network sample rate SR and the reconstruction measurement rate MR in the range of 0.1–0.9, and compared the reconstruction results with those of the traditional one-norm prior and TVAL3 to verify the feasibility of the structure, aiming to investigate the computational complexity of employing the sparse autoencoder prior for the single-photon reconstruction and the impact of network parameters on the reconstruction outcomes.

Secondly, to enhance the performance of the sparse autoencoder, we needed to adjust the sparsity parameter in the network. Therefore, we optimized the coefficient parameters for the networks trained on the MNIST and FACE datasets using the dichotomy method and then studied the λ prior ratio of the corresponding sparse autoencoder network according to the order of magnitude of fidelity term and prior term during the iteration trained on different datasets to explore the best prior reconstruction effect with a sum ratio.

Furthermore, we adjusted the network sample rate SR and reconstruction measurement rate MR of the sparse autoencoder for the MNIST and FACE datasets with adjusted parameters to investigate the effect of network depth on reconstruction results. We also used a multi-channel prior method to explore whether it has a better generalization ability and whether it can improve the prior reconstruction results.

Finally, we applied the principles and related data used in the simulation experiments to the single-photon image reconstruction system in the laboratory and conducted two experiments at low sampling rates. We selected four measurement rates (MR = 0.05, 0.1, 0.3, 0.4) and analyzed and discussed the results by comparing the peak signal-to-noise ratio (PSNR) with the original images. The ultimate goal of this study is to explore the reconstruction ability of this method for single-photon images.

The following formulas were used in the research process:

$$PSNR(x, \hat{x}) = 20 log_{10} \frac{Max(\hat{x})}{||x - \hat{x}||_2}, \tag{13}$$

assuming that $x \in \mathbb{R}^{mn}$ is the image to be reconstructed, and $\hat{x} \in \mathbb{R}^{mn}$ is the original image.

### 4.2. Experiment Result

4.2.1. Comparison between Sparse Autoencoder (SAE) Network Prior and Other Methods

The reconstruction measurement rate of the measurement matrix in $AX - Y$ is defined as MR, while the network sampling rate in the sparse autoencoder (SAE) network is defined as SR. Table 1 shows the reconstruction results of the SAE network prior, traditional one-norm prior reconstruction, and TVAL3 at the same dataset, changing MR when SR was fixed. The image reconstruction results are shown in Figure 7. Table 2 presents the complexity of the two components involved in this method, namely, the training parameter complexity of the sparse autoencoder and the computational complexity of the prior-constrained single-photon reconstruction algorithm.

The results show that the method of reconstructing images using SAE network prior information performs better than one-norm prior at most MR and is significantly competitive when MR is low compared with TVAL3, which is consistent with the experimental principles mentioned in Section 2.

**Table 1.** Reconstruction PSNR of different methods at different measurement rates MR.

| Prior | MR = 0.05 | MR = 0.1 | MR = 0.2 | MR = 0.3 | MR = 0.6 |
|---|---|---|---|---|---|
| TVAL3 | 10.82683 | 9.84631 | 15.41595 | 22.62640 | 44.17010 |
| one-norm | 15.60288 | 15.83556 | 16.57922 | 17.34535 | 20.72902 |
| SAE (0.05) | 15.63417 | 15.93219 | 16.71110 | 17.35355 | 19.37229 |
| SAE (0.6) | 15.59581 | 15.85697 | 16.78925 | 17.55059 | 20.41493 |

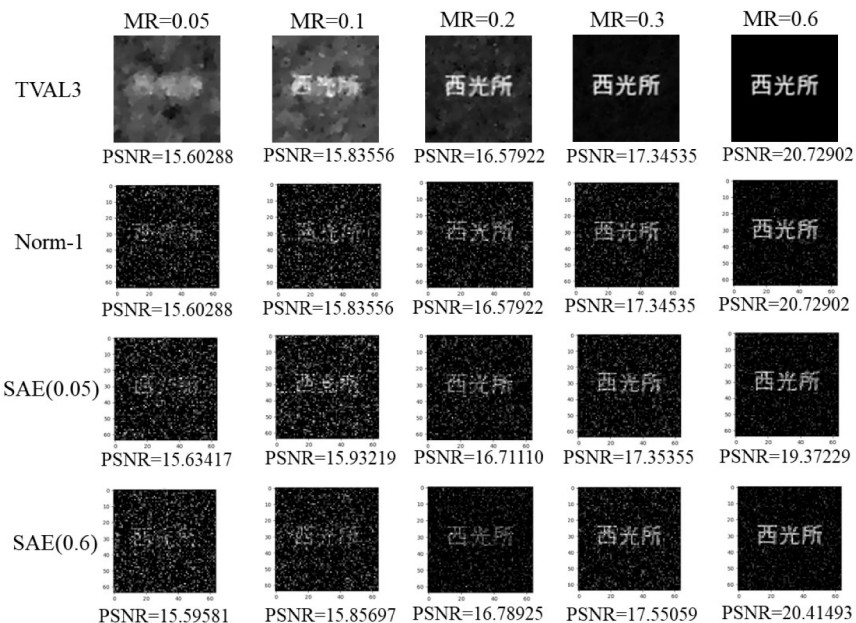

**Figure 7.** Reconstruction results with different methods (MNIST). We use the image with the Chinese characters of our affiliation on it and reconstruct it using TVAL3 and norm-1 prior as comparison and then test the SAE prior under different SR.

**Table 2.** Computational complexity of single-photon reconstruction based on SAE prior.

| Computational Complexity Estimation | Compute Complexity | Parameter Count | Training Epochs | Training Time (h) |
|---|---|---|---|---|
| SAE | 16.778 M | 16.788 M | 1000 | 0.025 |
| $\underset{x}{Min}\|\|Ax-y\|\|^2 + \lambda \times \|\|D(x)-x\|\|^2$ | Around 50,000 iterations | | 60 | 1.5 |
| $\underset{x}{Min}\|\|Ax-y\|\|^2 + \lambda \times \|\|x\|\|_1$ | Around 50,000 iterations | | 80 | 2 |

Additionally, considering the computational complexity of the two components of this approach, it can be observed that the first part, involving the sparse autoencoder network, offers advantages such as lightweight structure, fast reconstruction, and high-quality reconstructed images. This part can be trained quickly and achieve excellent results. The computational complexity of the second part, which involves single-photon reconstruction controlled by the prior constraint, is estimated using gradient descent. It assumes that the required number of computations for each operation and the amount of data involved are proportional and uses a multiplication factor to estimate the computational complexity of each operation. The computational complexity for each epoch operation is recorded as 50,000 computations. Table 3 shows that the sparse autoencoder (SAE) prior involves approximately 20 more epochs compared to the one-norm prior. Therefore, it is concluded that the image reconstruction effectiveness of using a sparse autoencoder network as prior information surpasses that of the one-norm prior reconstruction method without significantly sacrificing computational complexity to achieve performance improvement. Thus, it can be deemed advantageous to employ a sparse autoencoder network as prior information.

**Table 3.** Training results at different Rho_target values with SR = 0.5 and beta = 0.1 (MNIST).

| Rho_t | 0.01 | 0.05 | 0.055 | 0.056 | 0.1 |
|---|---|---|---|---|---|
| PSNR | 20.27770 | 20.43110 | 20.52952 | 20.50590 | 20.41738 |

We investigated the reconstruction performance of the SAE prior in low (SR = 0.05) and high (SR = 0.6) sampling rates. Compared with the results using one-norm prior, the advantage of the SAE prior became more prominent when the sampling rate SR and the prior measurement rate MR were both low. Correspondingly, when the sampling rate SR and the prior measurement rate MR were both high, the advantage of the SAE prior became more remarkable. We believe that this is due to the powerful feature extraction ability of the SAE because when reconstructing the prior information using this network, it has a stronger image reconstruction ability compared with adopting the one-norm prior.

In this experiment, we found that the reconstruction results of the SAE prior were better when the SR and MR were matched than when they were not. We realized that this was because the features of the image we use were similar to those of the MNIST dataset and had fewer features. Therefore, the SAE prior with SR = 0.05 and MR = 0.05 had a stronger capability of capturing the main image features, while the SAE prior with SR = 0.6 might capture redundant features and, thus, had worse performance than that with SR = 0.05. This suggests that it is more favorable to have both SR and MR low for reconstruction. Similarly, when MR = 0.6, the SAE prior with SR = 0.6 could better satisfy the feature capturing requirements; that is, the features captured by the SAE prior network conformed to the reconstruction requirements of $AX - Y$. Therefore, the effect was better when both SR and MR were high than when MR was high and SR was low.

### 4.2.2. Select the Optimal SAE Network Parameters and Prior Reconstruction Parameter λ

To enhance the performance of the prior network during the reconstruction process, it is necessary to adjust the parameters of both the SAE network and the prior reconstruction. Owing to the varying number of image features present across different datasets and other reasons, the FACE dataset has more complex features than the MNIST dataset; we have carried out distinct parameter optimization procedures for each of these datasets.

1.  Parameters of SAE Network

The SAE network adds constraints to the loss function compared to the traditional autoencoder. Among them, there is a control parameter of the hidden layer state, the expected sparsity ratio Rho_t, used to represent the expected average activation of the hidden layer, thus affecting the compression performance and generalization ability of the model, as well as the coefficient penalty term beta. The higher the value of beta, the higher the sparsity requirement of the encoder's hidden layer, which may result in better compression and generalization performance but, correspondingly, may also make network training more difficult.

In this experiment, we fixed beta and Rho_t and trained the network at the same SR to identify the optimal parameters for the SAE network on the MNIST and FACE datasets. The process is shown in Figure 8.

For the MNIST dataset, we first measured the PSNR under different Rho_t with beta = 0.1. The experimental results show that the network training achieves optimal performance when the expected sparsity ratio Rho_t is around 0.055. Next, we measured the PSNR under different beta with fixed Rho_t = 0.05, and the experimental results show that the sparsity coefficient penalty term achieves optimal performance when it is around 0.2; that is, the network can perform better with Rho_t = 0.055 and beta = 0.2. Then, we measured the optimal parameters of the FACE dataset in the same way as above, showing that the network trained by the FACE dataset performs better when Rho_t = 0.01 and beta = 0.3. The results are shown in Tables 3–6.

**Table 4.** Training results at different beta values with SR = 0.5 and Rho_t = 0.05 (MNIST).

| Beta | 0.1 | 0.15 | 0.2 | 0.23 | 0.25 |
|------|-----|------|-----|------|------|
| PSNR | 20.43130 | 20.44929 | 20.48450 | 20.44459 | 20.43981 |

**Table 5.** Training results at different Rho_target values with SR = 0.5 and beta = 0.3 (FACE).

| Rho_t | 0.005 | 0.01 | 0.015 | 0.02 | 0.05 |
|---|---|---|---|---|---|
| PSNR | 18.67160 | 19.23764 | 18.67760 | 19.15409 | 18.95618 |

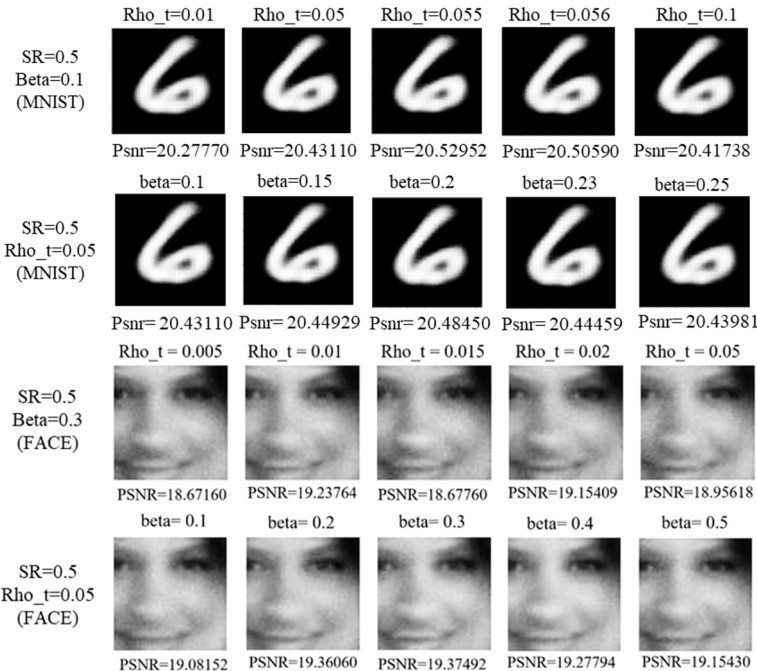

**Figure 8.** Reconstruction results with different parameters using different datasets. We used MNIST and FACE datasets to train the SAE network and test the best performance parameters Rho_t and beta separately.

**Table 6.** Training results at different beta values with SR = 0.5 and Rho_t = 0.05 (FACE).

| Beta | 0.1 | 0.2 | 0.3 | 0.4 | 0.5 |
|---|---|---|---|---|---|
| PSNR | 19.08152 | 19.36060 | 19.37492 | 19.27794 | 19.15430 |

2. The prior balance parameter $\lambda$ in the prior reconstruction

The prior balancing parameter $\lambda$ in the reconstruction objective balances the impact of the compressive sensing fidelity term and the prior constraint. If its value is excessively large, it could undermine the effect of the fidelity term, while its small value would render the prior effect less pronounced. Hence, by analyzing the relative scale and gradient changes of the two during training, we could determine the optimal value. Moreover, as different image features exist across various datasets, the magnitudes of training quantities also vary, implying that SAE prior networks based on different datasets should exhibit distinct $\lambda$ values, and the process is shown in Figure 9.

In our experiment, we conducted a $\lambda$ test based on the MNIST dataset under the condition of SAE prior SR = 0.3, MR = 0.3, Rho_t = 0.055, and beta = 0.2. We selected a value range of 0.1–10 for the initial scale of the fidelity term and prior constraint during training and evaluated the reconstruction effects under various $\lambda$ values. The experimental results showed that the reconstruction quality was the best when $\lambda$ = 0.5, with the reconstructed images exhibiting higher PSNR, as presented in Table 7. During the training, with step_num = 1, the fidelity term result1 starts at a scale of $10^{10}$ and gradually decreases to $10^3$ with increasing training iterations. To enable the fidelity term and the prior to achieve a more synchronized gradient impact, we should select the $\lambda$ value that allows for the prior

constraint result2 to fit into this range. When λ = 0.5, result2 is at the scale of $10^6$, meeting our expectations.

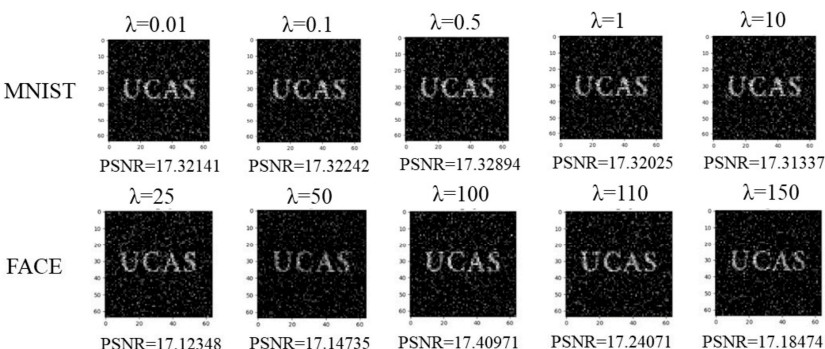

**Figure 9.** Reconstruction results with different λ using different datasets. We used MNIST and FACE datasets to train the SAE network and test the best performance parameter λ.

**Table 7.** Reconstruction results at different values of λ (MNIST).

| λ | 0.01 | 0.1 | 0.5 | 1 | 10 |
|---|---|---|---|---|---|
| PSNR | 17.32141 | 17.32242 | 17.32894 | 17.32025 | 17.31337 |

Similarly, these experiments were conducted on the FACE dataset under the conditions of SR = 0.3, MR = 0.3, Rho_t = 0.01, and beta = 0.3 to examine the reconstruction effects under different λ values. The result shows that the best λ under the FACE dataset is 100, as presented in Table 8.

**Table 8.** Reconstruction results at different values of λ (FACE).

| λ | 25 | 50 | 100 | 110 | 150 |
|---|---|---|---|---|---|
| PSNR | 17.11100 | 17.12348 | 17.14735 | 17.40971 | 17.24071 |

4.2.3. The Effects of Sampling Rate SR and Reconstruction Measurement Rate MR of SAE Prior on Image Reconstruction of Different Datasets

To investigate the impact of different datasets as prior information for SAE on reconstruction, the same network structure was used to train the MNIST dataset and the FACE dataset separately. The prior results obtained from training were then used for the reconstruction of different test images.

Based on the experimental results presented in our previous study, we reconstructed the image "UCAS" using the optimal reconstruction parameters obtained from the SAE prior and the reconstruction process. Figures 10 and 11 illustrate the reconstructed results. Our experimental results demonstrate that changing the sampling rate SR while keeping the measurement rate MR constant has little effect on the overall reconstruction quality. However, changing the measurement rate MR while keeping the sampling rate SR constant has a significant impact on the reconstruction quality. As "UCAS" has a relatively low number of features, it still maintains some reconstruction competitiveness even under low sampling rates. The results are shown in Tables 9 and 10.

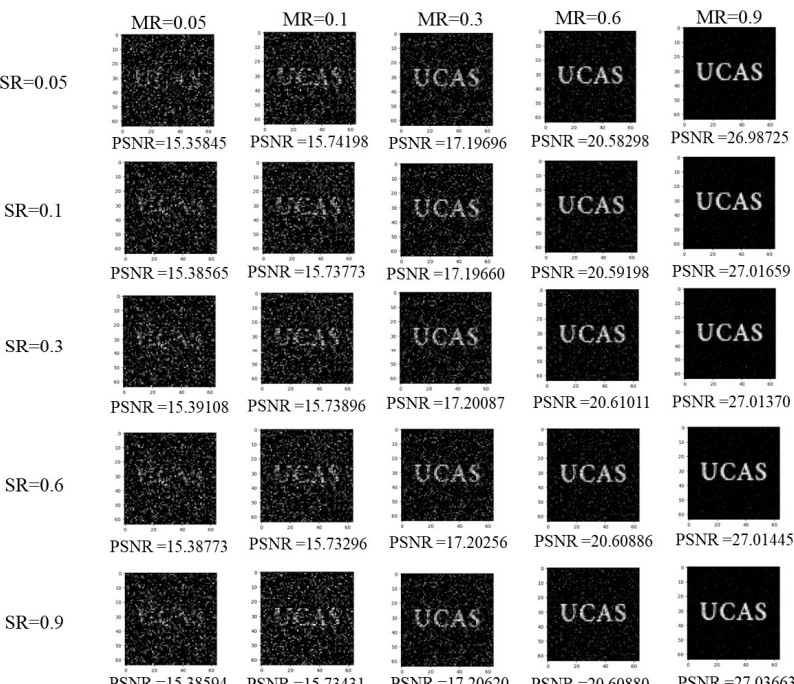

**Figure 10.** Reconstruction results at different values of SR and MR (MNIST).

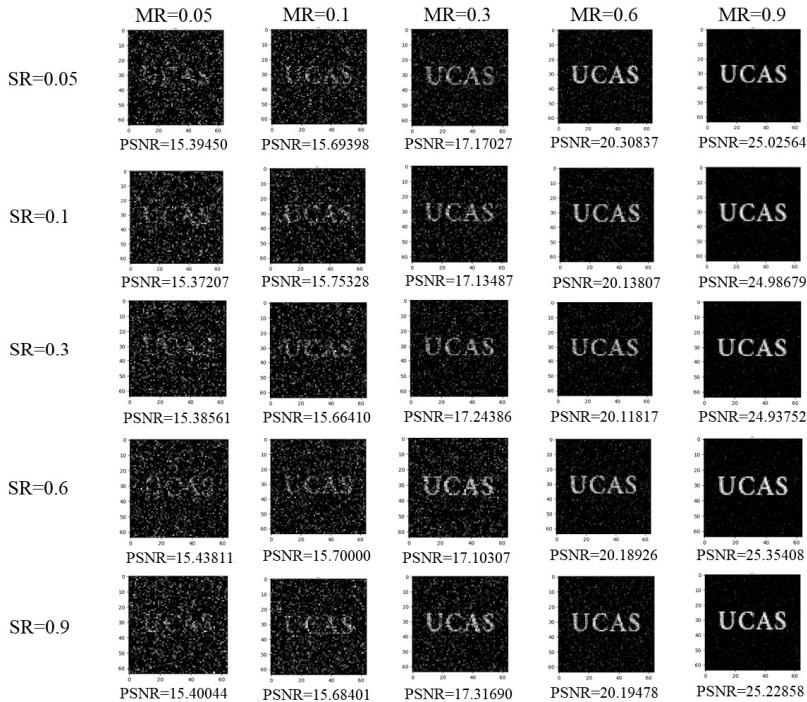

**Figure 11.** Reconstruction results at different values of SR and MR (FACE).

**Table 9.** Reconstruction results at different values of SR and MR (MNIST).

| SR | MR = 0.05 | MR = 0.1 | MR = 0.3 | MR = 0.6 | MR = 0.9 |
|---|---|---|---|---|---|
| 0.05 | 15.35845 | 15.74198 | 17.19696 | 20.58298 | 26.98725 |
| 0.1 | 15.38565 | 15.73773 | 17.19660 | 20.59198 | 27.01659 |
| 0.3 | 15.39108 | 15.73896 | 17.20087 | 20.61011 | 27.01370 |
| 0.6 | 15.38773 | 15.73296 | 17.20256 | 20.60886 | 27.01445 |
| 0.9 | 15.38594 | 15.73431 | 17.20620 | 20.60880 | 27.03663 |

**Table 10.** Reconstruction results at different values of SR and MR (FACE).

| SR | MR = 0.05 | MR = 0.1 | MR = 0.3 | MR = 0.6 | MR = 0.9 |
|---|---|---|---|---|---|
| 0.05 | 15.39450 | 15.69398 | 17.17027 | 20.30837 | 25.02564 |
| 0.1 | 15.37207 | 15.75328 | 17.13487 | 20.13807 | 24.98679 |
| 0.3 | 15.38561 | 15.66410 | 17.24386 | 20.11817 | 24.93752 |
| 0.6 | 15.43811 | 15.70000 | 17.10307 | 20.18926 | 25.35408 |
| 0.9 | 15.40044 | 15.68401 | 17.31690 | 20.19478 | 25.22858 |

### 4.2.4. The Effect of Multi-Channel Prior Experiment on Prior Reconstruction

Based on the conclusion in Section 4.2.1, we understand that the low MR prior reconstruction has better performance than the one-norm prior when the SR is low, while the high MR prior reconstruction has better performance than the one-norm prior when the SR is high. Therefore, we aim to improve the model's generalization ability by combining the low SR and high SR channels and incorporating fidelity terms. We adopt a multi-channel approach to incorporate the autoencoding prior, selecting appropriate weight parameters, and minimizing the following expression:

$$J(x) = \underset{x}{Min}||Ax - y||^2 + \lambda_1 \times ||D_1(x) - x||^2 + \lambda_2 \times ||D_2(x) - x||^2, \tag{14}$$

$||D_1(x) - x||^2$ and $||D_2(x) - x||^2$ represent the prior information of two autoencoder networks. In this experiment, SR = 0.05 and SR = 0.9 channels were selected for comparison and analysis, combined with a single SR = 0.05 and SR = 0.9 channel.

The experimental results in Figure 12 demonstrate that the multi-channel SAE network effectively addresses the drawbacks of the single-channel method while preserving its advantages, resulting in good performance for reconstructing low and high-measurement-rate images. This validates the hypothesis that multi-channel approaches can enhance the performance of prior-based image reconstruction. This trade-off approach effectively balances the strengths of both SR = 0.05 and SR = 0.9 scenarios, leading to improved prior-based image reconstruction. The findings suggest that the multi-channel SAE network can be a promising approach for enhancing the performance of prior-based image reconstruction. However, this will significantly increase the difficulty of adjusting parameters, which is a time-consuming and laborious task.

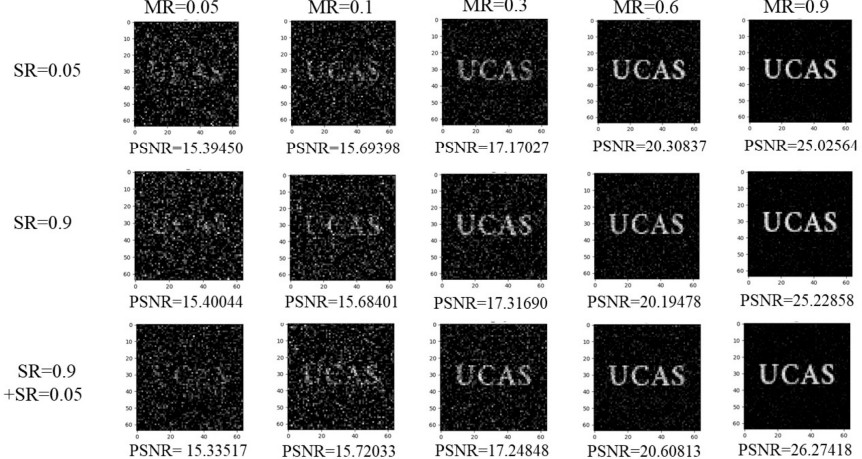

**Figure 12.** Reconstruction results with multi-channel (FACE).

### 4.2.5. Application of Network Prior in Single-Photon Image Reconstruction Results in Laboratory Experiments

Application of $\pm 1$ binary matrix used in simulation training in a single-photon imaging system in the laboratory. Applications 1 and $-1$ are converted to 0, respectively, and

the subtraction of the measurement values applied on DMD is used as the actual measurement value Y. MR = 0.05, 0.1, 0.3, 0.4 are selected as the low measurement rate, and three images are reconstructed. Specifically, it should be noted that the mask on the template is slightly different from the actual original image in terms of angle, scaling, and padding. In order to ensure the rigor of this experiment, the well-reconstructed image at a sampling rate of 0.9 is used as the input image for prior reconstruction, and the reconstructed images are compared with the input image by calculating the PSNR.

The reconstruction results are shown in Figures 13 and 14, and the gradient during this experiment is shown in Figure 15. In addition, when comparing the results of Figures 13 and 14 with those of Figures 10 and 11, we found that the reconstruction results are better at lower MR when the actual measurement value Y of single-photon reconstruction is applied to binary matrix reconstruction.

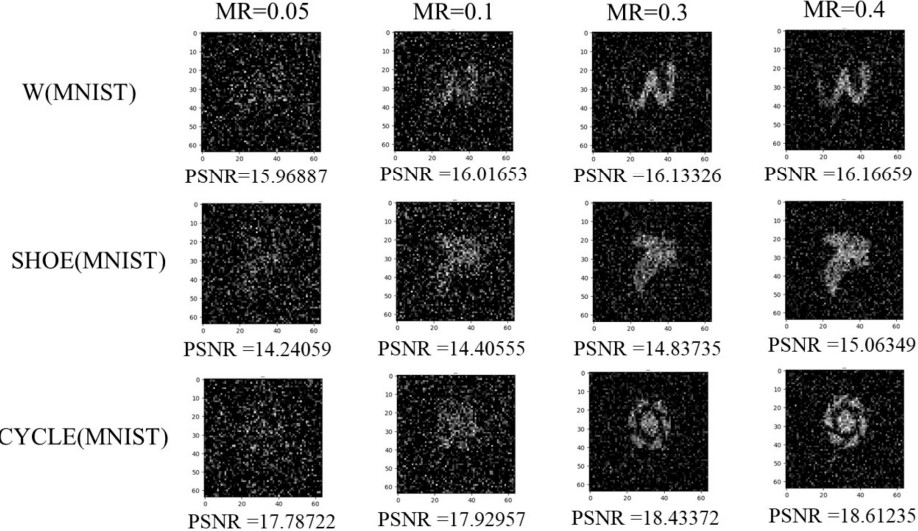

**Figure 13.** The reconstruction effect in laboratory experiments (MNIST).

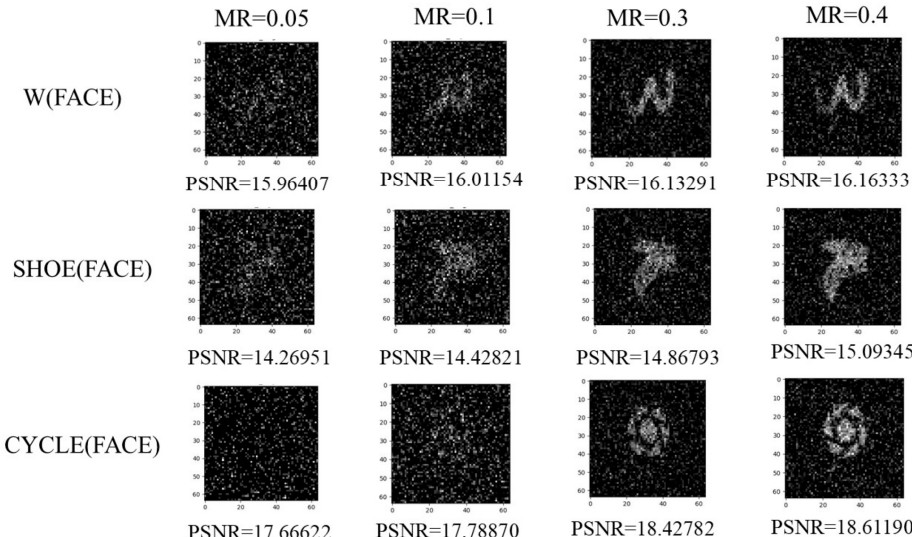

**Figure 14.** The reconstruction effect in laboratory experiments (FACE).

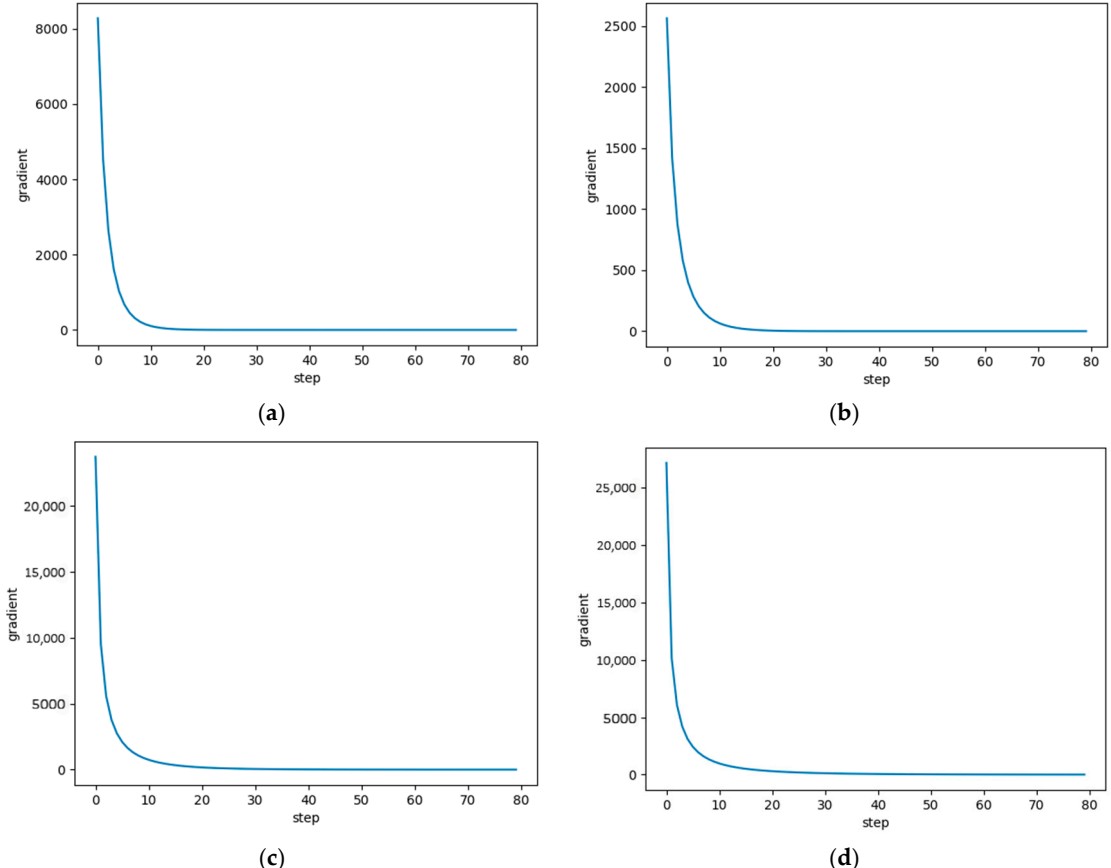

**Figure 15.** The gradient results of different MRs in laboratory experiments reconstruction process. We recorded the gradient data under different MRs and plotted them every 5 steps. (**a**) The result of MR = 0.05. (**b**) The result of MR = 0.1. (**c**) The result of MR = 0.3. (**d**) The result of MR = 0.4.

Although the reconstruction effect was slightly worse at MR = 0.3 compared to the previous simulation experiment, the prior reconstruction method had a better reconstruction effect at lower MR. This indicates that the SAE prior reconstruction method is more competitive at low measurement matrix sampling rates and performs better in real experiments at low sampling rates than in simulation experiments.

## 5. Conclusions

We propose a new sparse autoencoder network prior for the single-pixel image reconstruction. Unlike traditional prior information, this method uses the trained results of the sparse autoencoder network as the prior for the image reconstruction, and we demonstrate that SAE prior transforms sparsity constraints of the prior terms into contour similarity constraints, effectively optimizing the role of prior terms in the reconstruction process. Experimental results show that this method is more suitable for images with distinct features, such as MNIST images and FACE images, and has significant advantages in reconstructing single-photon images. Compared with the traditional one-norm prior and TVAL3, the proposed sparse autoencoder network prior outperforms the traditional one-norm prior when using matching prior sampling rates and reconstruction measurement rates in single-channel prior, and the reconstruction quality of the multi-channel prior is superior to one-norm prior at any measurement rates when adjusting the weight settings, effectively improving the reconstruction quality and flexibility while maintaining the cost of computation at a similar level with one-norm prior. Furthermore, the performance of our SAE prior is significantly competitive when MR is low compared to the TVAL3. Therefore, it is suitable for various scenarios and has broad development prospects. However,

the disadvantage is that it has poor generalization ability and is unable to effectively reconstruct natural images with less obvious features under low sampling rates.

**Author Contributions:** Conceptualization, W.C. and H.W.; funding acquisition, H.W.; investigation, S.D. and W.C.; methodology, H.Z. and H.G.; software, H.Z., J.D. and Q.L.; writing—original draft, H.Z.; writing—review and editing, J.D., Q.L. and H.W.; visualization, S.D. and H.G. All authors have read and agreed to the published version of the manuscript.

**Funding:** This research was funded by the West Light Foundation of the Chinese Academy of Sciences, grant No. XAB2021YN15.

**Institutional Review Board Statement:** Not applicable.

**Informed Consent Statement:** Not applicable.

**Data Availability Statement:** The data underlying the results presented in this paper are not publicly available at this time but may be obtained from the authors upon reasonable request.

**Conflicts of Interest:** The authors declare no conflict of interest.

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
