# Peer review of "Compressive Reconstruction Based on Sparse Autoencoder Network Prior for Single-Pixel Imaging"

_photonics, doi:10.3390/photonics10101109_

Round 1
Reviewer 1 Report (Previous Reviewer 2)
I find the authors have revised the manuscript, with the presentation quality improved. I think it is almost ready for publication, with some further revision.
1. More details about the autoencoder are expected. Such as, what kind of specific features of images are considered? What kind of loss function is used?
2. I sugguest the authors put some details about the figure in the caption, to make pictures more instructive.
3. line 72, Hadamard matrix is not a random matrix.
English is ok.
Author Response
Please see the attachment.

Reviewer 2 Report (Previous Reviewer 1)
This manuscript has been modified accordingly to the issues mentioned earlier. It is well revised.
However, there is a tense error in line 547:” However, the difficulty of adjusting the parameters increased, which will be a time-consuming and laborious work.”
Only after this issue is addressed will it be accepted.
the language needs a little recheck.
Author Response
Please see the attachment.

This manuscript is a resubmission of an earlier submission. The following is a list of the peer review reports and author responses from that submission.
Round 1
Reviewer 1 Report
This manuscript comes up with a compressed sensing reconstruction method based on sparse autoencoder network prior that can reconstruct the results of single photon counting compressed sensing imaging directly. And also a multi-channel prior information in image reconstruction is proposed too. Those methods improve the quality of the reconstruction.
However, there are some issues that need to be solved and corrected, as following:
Firstly, the experiment is well organized basing on single photon counting compressive imaging system. For a better understanding of the system and each instrument, in Figure 1 ,DMD needs to be marked. About the result, there are some errors : in chapter 4, table 2. The measured rate values in the first row are the same. In Figure 11 , wrong words :“Mr”.
Secondly, using concise words, this manuscript is readily comprehensible. However, there is an error in row 39, ”The imaging sensitivity of a system that is no longer limited by the detection sensitivity and imaging sensitivity of a single photon point detector, which can be very high.” A double-check is recommended.
Finally, about the content:
1.Basing on the SAE, this new prior constraint in row 206: “the squared magnitude of the autoencoder error” , a more detailed explanation on why this prior can be used theoretically is recommended.
2.In the part of SAE, comparing to simple and traditional structure, what works have done on optimizing SAE method? Here is not clear and needs a summary.
3.Besides, more additional constraints and prior knowledge proposed early need to be supplemented in the part of introduction, and also their disadvantages and advantages.
Only after these issues above are addressed will this paper be accepted.
A rechecking on the language is recommended.
Reviewer 2 Report
The authors discussed on compressive imaging based on sparse autoencoder network prior, which might be useful for the community. However, I think the manuscript is not ready for publication based on the following issues.
1. I failed to find out their discussion to be really related to single photon counting, different from the title. How the simulations and experiments were carried out considering single photon? What about the source, the detection? How did you deal with the probabilty issue of detection? What kind of data do you assumed to obtain from experiments? Number of photons?
2. line 59-72, the authors claimed that there are two different methods. However, I failed to understand how were thosed methods classified and what's the different.
3. When discussing on their compressive sensing reconstruciton method, they are comparing with that using 1-norm prior. What about other developped methods? Why this method is novel?
4. When using multi-channel prior, what should we pay for better performance?
5. There are many parameters to be determined before the method to be useful, how to do that?
6. What about the cost of computation?
ok
